# Trend Analysis of Patient Safety Incidents and Their Associated Factors in Korea Using National Patient Safety Report Data (2017~2019)

**DOI:** 10.3390/ijerph18168482

**Published:** 2021-08-11

**Authors:** Sunhwa Shin, Mihwa Won

**Affiliations:** 1College of Nursing, Sahmyook University, Seoul 01795, Korea; shinsh@syu.ac.kr; 2Department of Nursing, Wonkwang University, Iksan 54538, Korea

**Keywords:** patient safety, medical errors, public reporting of healthcare data, safety management

## Abstract

This study analyzed trends in patient safety incidents (PSIs) and the factors associated with the PSIs by analyzing 2017–2019 Patient Safety Report data in Korea. We extracted 2940 records in 2017, 5889 in 2018, and 7386 in 2019, from hospitals with more than 200 beds, and used all 16,215 cases for analysis. SPSS 25.0 was used for a multi-nominal logistic regression analysis. The PSI trend analysis, the standardized Jonckheere–Terpstra test was significant. On analyzing the probability of adverse events based on near misses, the significant variables were patient age, the season when PSIs occurred, incident reporter, hospital size, the location of PSIs, the type of PSIs, and medical department. Additionally, the factors that were likely to precipitate sentinel events based on near misses were patient sex, patient age, incident reporter, the type of PSIs, and medical department. To prevent sentinel events in PSIs, female and older patients are required to pay close attention. Moreover, it is necessary to establish a patient safety reporting system in which not only all medical personnel, but also patients, generally, can actively participate in patient safety activities and report voluntarily.

## 1. Introduction

Patient safety incidents (PSIs) are unintended or unexpected events that may bring unnecessary harm to a patient and are a significant clinical problem worldwide [1,2]. Recently, rising healthcare labor costs, advances in treatments, health technologies, and excessive information have increased PSI prevalence rates [3]. In Organisation for Economic Co-operation and Development (OECD) countries, patient harm may account for more than 6% of hospital bed days and more than 7 million admissions [4]. PSIs are reported to be the third leading cause of death in the USA [2]. A European Commission survey found that 27% of European Union citizens reported that they, or their family members, had experienced an adverse event while receiving healthcare [5]. In Korea, the prevalence rate of PSIs has gradually increased annually to 3864 in 2017, 9250 in 2018, and 11,953 in 2019 [4].

Particularly, a national review report on medical charts estimates that approximately 10% of hospital admissions are associated with adverse events, such as injuries resulting in prolonged hospitalization, disability, or death, caused during the healthcare process [6]. The prevalence of preventable patient harm was 6%, while 12% of preventable patient harm was severe or led to death [7].

Harmful patient incidents are also a major financial burden for healthcare systems globally [8]. Moreover, PSIs can lead to low confidence in the healthcare professions and prolonged hospital stays, as well as higher morbidity and mortality in healthcare settings [9,10]. Thus, it is important to identify PSI factors in clinical settings and decrease PSI rates, which are sensitive indicators of healthcare quality and patient cost.

A recent study shows an increased rate of medication-related no-harm incidents in the age group of 69–70 years [11]. A systematic review found that the prevalence rate of overall PSIs was 12%, and that patient events were associated with medication, invasive medical and surgical procedures as well as falls [12]. Moreover, PSIs were associated with drugs (25%), while other treatments (24%) accounted for the largest proportion of preventable patient harm in a meta-analysis [6]. A study by Kuo et al. [13] found that, compared to hospitals with PSIs, those without any PSIs showed significant differences in costs and the cost and length of stay for tube and line, and medication events. Additionally, some researchers reported that elderly inpatients are three times more likely to fall [14] and approximately 10% of elderly inpatient falls lead to sentinel events, such as mortality [15].

Currently, most advanced countries have not only introduced patient safety reporting systems to prevent the recurrence of adverse events, but have also enacted relevant legislation to establish a patient safety management system [16]. In Korea, the Korea Patient Safety reporting and learning system (KOPS) was introduced to collect systematic data on PSIs with the implementation of the Patient Safety Act in 2016 [4]. This system collects data on PSIs at the national level through voluntary reporting of patient safety by medical institutions and patients [4]. Thus, it analyzes the underlying causes of severe PSIs that may occur in patients and provides useful data on the prevalence of medical errors and its causes, based on a case record review [4]. Moreover, multidimensional efforts have been made based on the perspective of medical and healthcare providers to prevent medical malpractices, medical errors, and adverse events, by fostering a patient safety culture and implementing a patient safety reporting system [17]. Incident reports for patient safety in healthcare are considered significant in enhancing patient safety strategies, monitoring, and prevention, as well as in reducing the occurrence of patient safety events [18]. Thus, the reporting of safety events is important for the broad goal of error reduction [19]. Despite the importance of PSIs in clinical settings, most medical institutions focus on acute treatment, so there is a lack of PSI reports in Korea [20]. Also, the awareness of patient safety problems and evidence for prevention remains low in the country; thus, it is estimated that the frequency of PSI experiences is lower than in other countries [20]. To date, few studies have been published on the frequency or proportion of PSIs based on medical record reviews [21,22]. Despite the high rate of PSIs worldwide, including among the Korean population, the characteristics associated with the occurrence of PSIs in general hospital settings have not been established using national data. Therefore, this study aims to analyze the PSI trends and their associated factors using national Patient Safety Report data from 2017 to 2019.

## 2. Materials and Methods

### 2.1. Study Design and Sample

This study involved a secondary data analysis based on raw data from 2017 to 2019, collected for the Korea Patient Safety Report survey, a nationally representative survey conducted by the Korea Institute for Healthcare Accreditation. The KOPS was established for the safety of medical institutions; it collects information on patient safety at the national level [4]. The system analyzes the root causes of serious incidents that may occur in relation to patients and provides useful data on the prevalence and causes of medical errors through case record review [19,23].

The survey data from 2017 to 2019 involved 3864 reported PSIs in 2017, 9250 in 2018, and 11,953 in 2019 [4]. For this study, data from general hospitals with more than 200 beds were extracted for analysis, comprising 2940 cases in 2017, 5889 in 2018, and 7386 cases in 2019, summing up to 16,215 cases.

### 2.2. Ethical Considerations

The data source, the Patient Safety Report’s Learning System, is publicly provided by the Korea Institute for Healthcare Accreditation in such a manner that participants cannot be identified. The institutional review board at Sahmyook University exempted this study from their board review (IRB No: 2-1040781-A-N-012020088HR).

### 2.3. Research Variables

#### 2.3.1. Severity of PSIs

The severity of PCIs was classified as near miss, adverse, and sentinel events, based on previous studies [13,24,25]. A near miss is defined as a preventable event in which a situation that could cause injury to a patient occurs, but leads to no actual injury [26,27]. An adverse event means that a harm is caused to the patient by medical treatment or healthcare problems, rather than by the patient’s current disease [28]. A sentinel event is defined as the most serious event, with death or serious physical or mental damage, or with unexpected consequences [29].

#### 2.3.2. Type of PSIs

The PSIs were classified into different types: falls, medication/transfusion, surgery/anesthesia/examination, infection/contamination, equipment/computational disorder, and others. Others included self-extubation (intubation or drainage tube, etc.), suicide/self-harm, escape/disappearance, violence/riot, burns, sore, etc., and medical department (internal medicine, surgical, complex support division, or clinical support division).

#### 2.3.3. Covariables

The covariables in our study include patient sex, patient age, the time of occurrence of PSIs (day, evening, or night), the season of occurrence of PSIs (spring, summer, fall, or winter), incident reporter (health provider or patient safety officer), hospital size (hospitals with 200 to 499 beds, and those with over 500 beds), the location of occurrence of the PSIs (general ward, outpatient clinic, special ward, or others), the type of PSIs, and the severity of PSIs.

### 2.4. Statistical Analysis

For data analysis, the SPSS/WIN 25.0 (SPSS, IBM Armonk, NY, USA) program was used. For the participants’ demographic characteristics, the descriptive statistics of frequency, percentage, mean, and standard deviation were analyzed. The difference in PSIs based on the demographic characteristics by year was cross-analyzed using the χ^2^ test, while the Jonckheere-Terpstra (J-T) test was performed to analyze the trend of PSIs by year. A multinomial logistic regression analysis was performed to identify the factors related to the PSIs (near miss, adverse, and sentinel events). Regression Model I analyzed the probability (odds ratio, OR) of an adverse event using near miss as a reference category, while Regression Model II analyzed the probability (OR) of a sentinel event using near miss as a reference category. The results are presented as ORs with 95% confidence intervals (CIs), with *p* < 0.05 as the level of statistical significance.

## 3. Results

### 3.1. Trend by Severity of PSIs

The three-year trend by the severity of the PSIs was statistically significant (χ^2^ = 171.16, *p* < 0.001). Regarding the severity of PSIs, the prevalence of both adverse and sentinel events gradually decreased, while the prevalence of near miss events gradually increased (Figure 1).

### 3.2. Difference in PSIs Severity by Research Variables

The male (49.8%) and female (50.2%) patients had similar PSIs rates, while 68.0% of all the patients were aged 60 years and older. Regarding the time of occurrence, 48.9% of the PCIs occurred during the day. The highest proportion of PSIs occurred in summer (28.6%), followed by spring (25.1%), fall (24.3%), and winter (22.0%). Most incidents were reported by a patient safety officer (90.6%), while hospital sizes between 200 and 499 beds accounted for 50.1% of the sample. Regarding the location of incidents, most PSIs occurred in general wards (56.0%). In terms of the type of PSIs, most incidents were falls (56.4%) (Table 1).

In medical departments, the highest percentage of PSIs occurred with internal medicine (59.2%), followed by surgical division (30.4%), complex support division (6.8%), and clinical support division (3.7%). The severity of PSIs was statistically different between the sexes (χ^2^ = 96.93, *p* < 0.001) and patient ages (χ^2^ = 230.67, *p* < 0.001). For the remainder of the variables, the following results were obtained: time when the PSIs occurred (χ^2^ = 66.77, *p* < 0.001), incident reporter (χ^2^ = 67.81, *p* < 0.001), hospital size (χ^2^ = 62.81, *p* < 0.001), the location of occurrence of PSIs (χ^2^ = 133.21, *p* < 0.001), type of PSIs (χ^2^ = 1203.62, *p* < 0.001), and medical department (χ^2^ = 136.74, *p* < 0.001) (Table 1).

### 3.3. Influence Factors Related to the Severity of PSIs

Table 2 presents the results from the multinomial logistic regression examining the influence of the variables on the severity of PSIs.

First, Regression Model I calculated the OR of an adverse event using near miss as a reference category. Patient age, the seasons when PSIs occurred, incident reporter, hospital size, the location of occurrence of PSIs, the type of PSIs, and medical department were significantly associated with an adverse event.

In terms of age, compared to patients below 19 years old, those between 20 and 59 years old (Adjusted OR (aOR) = 1.23, 95% CI = 1.05–1.43) and those 60 years old or older (aOR = 1.35, 95% CI = 1.17–1.57) had a significantly higher probability of experiencing an adverse event. Regarding the seasons when PSIs occurred, compared to those in winter, the incidents occurring in spring (aOR = 1.15, 95% CI = 1.04–1.27), summer (aOR = 1.12, 95% CI = 1.01–1.23), and autumn (aOR = 1.11, 95% CI = 1.00–1.23) were significantly more likely to be classified as adverse events.

A health provider was significantly more likely to report adverse events than a patient safety officer (aOR = 1.53, 95% CI = 1.35, 1.74). Adverse events were significantly more likely to occur in hospitals with more than 500 beds than in those with 200 to 499 beds (aOR = 1.22, 95% CI = 1.13–1.31). In terms of the location of occurrence of PSIs, adverse events were significantly more likely to occur in general wards (aOR = 1.23, 95% CI = 1.12–1.34), outpatient clinics (aOR = 1.54, 95% CI = 1.32–1.78), and special wards (aOR = 1.65, 95% CI = 1.42–1.91), than in “other” locations.

For the types of PSIs, compared to medication/transfusion, falls (aOR = 2.73, 95% CI = 2.47, 3.01), infection/contamination (aOR = 1.64, 95% CI = 1.34–2.02), and others (aOR = 3.81, 95% CI = 3.33–4.37) showed a higher probability, while surgery/anesthesia/examination had a significantly lower probability (aOR = 0.75, 95% CI = 0.65–0.86). Regarding medical departments, adverse events were more likely to occur in internal medicine (aOR = 1.42, 95% CI = 1.23–1.64), surgery (aOR = 1.26, 95% CI = 1.09–1.46), and clinical support departments (aOR = 1.92, 95% CI = 1.53–2.42), than in complex support divisions.

Second, Regression Model II analyzed the probability of a sentinel event using near miss as a reference category. Sentinel events were significantly associated with gender, age, incident reporter, the location of PSIs, and medical department.

The probability that females would experience a sentinel event was significantly higher than that for males (aOR = 1.75, 95% CI = 1.54–1.98). The probabilities of sentinel events occurring among those aged 20 to 59 years (aOR = 2.97, 95% CI = 1.80–4.90) and those over 60 years old (aOR = 5.02, 95% CI = 3.08–8.19) were significantly higher than the probability for those below 19 years old. Health providers were significantly more likely to report sentinel events than patient safety officers (aOR = 2.23, 95% CI = 1.85–2.70). The location of sentinel events was significantly more likely to be special wards than the reference category of others (aOR = 1.40, 95% CI = 1.05–1.87).

Sentinel events were significantly more likely to occur in falls (aOR = 11.83, 95% CI = 8.58–16.31), infections/contamination (aOR = 2.13, 95% CI = 1.05–4.33), surgery/anesthesia/examinations (aOR = 2.80, 95% CI = 1.85–4.22), equipment/computational failure (aOR = 4.01, 95% CI = 1.93–8.34), and others (aOR = 25.49, 95% CI = 18.10–35.09) than during medication/transfusion. Among the types of PSIs, the others type was found to be 25.49 times more likely to include sentinel events than medication/transfusion.

Regarding medical departments, with the combined support department as a reference category, the departments of internal medicine (aOR = 2.01, 95% CI = 1.50–2.79), surgery (aOR = 1.45, 95% CI = 1.05–2.00), and clinical support (aOR = 1.89, 95 % CI = 1.08–3.30) showed a significantly higher probability of experiencing sentinel events.

## 4. Discussion

Our study was conducted to identify factors related to PSIs based on the Korea Patient Safety Report data survey from 2017 to 2019. The three-year PSIs trend by severity was statistically significant, the prevalence of adverse and sentinel events decreased, while the prevalence of near misses increased. In this cross-sectional study, near misses accounted for 34.5% of the PSIs, adverse events 56.7%, and sentinel events 8.8%, indicating that there were fewer reports of near misses compared to previous studies. In contrast, a study by de Vries found the proportions to be 56.3% for near misses, 26.1% for adverse events, transient and permanent damage, and 7.4% for sentinel events, based on 74,485 patient records [29]. In Korea, the rate of near miss reports by general nurses was 63.2%, while the introduction of the healthcare accreditation system has increased awareness of the near miss report [30].

A near miss refers to an event that has prevented the occurrence of a PSI in advance and, as a result, does not cause harm to the patient, but may cause medical errors if it is not prevented [31,32]. Therefore, reporting a near miss is extremely important to prevent the occurrence of sentinel events, thus providing quality nursing care [30,33]. A previous study showed that nurses who were aware of near misses were more likely to report the PSIs than those who were not [30]. Therefore, it is necessary to provide training to healthcare providers to recognize the importance of the near miss report. Moreover, efforts are required to establish a culture of patient safety by creating an open and non-punishable atmosphere at the organizational level of medical institutions.

Regarding the type of PSIs, inpatient falls occurred most frequently (56.4%), followed by medication/transfusion. A previous study showed that 1 out of 10 patients experienced an adverse event, such as surgery error (39.6%) or medication error (15.1%) [29]. These results are different from the results of our study, which shows a large proportion of inpatient falls. In this study, the ORs of inpatient falls being classified as adverse events (2.73 times) and sentinel events (11.83 times) were higher than the OR for medication/transfusion errors. In a study analyzing types of patient safety events, based on the UK National Health Service Database, failure to act on deterioration in patients accounted for 23%, inpatient falls 10%, while healthcare-associated infections accounted for 10%. Based on these results, the risk of inpatient falls is high [34]. In our study, consistent with previous studies, inpatient falls showed a higher incidence of sentinel events than other PSIs. Therefore, healthcare providers should recognize the importance of management in inpatient falls. Sentinel events lead to unexpected and fatal results; therefore, healthcare providers are critical in preventing sentinel events in clinical settings. Furthermore, it is necessary for healthcare providers to be able to report PSIs and to find solutions; in addition, systematic education and training should be provided. In our study, the prevalence of sentinel events was higher among female patients and those aged 65 years and older. In previous studies, the probability of reporting PSIs among adults over 50 years of age was 2.29 times higher than among those under 50 years of age [19], while for the probability of adverse events occurring among people over 65 years old [35], the results were similar. Regarding hospital characteristics, the occurrence of adverse and sentinel events was high during night working hours, in hospitals with 200 to 499 beds, and in general wards. Therefore, it is necessary to report patient safety issues in a timely manner and to find solutions; meanwhile, systematic education and training should be provided.

In previous studies, small hospitals, with less than 200 beds, were 1.4 times more likely to experience adverse events than large hospitals, with 500 beds or more [35]. In addition, medical service had a higher incidence of adverse events than surgical service [35], while PSIs were reported more in general wards than in outpatient clinics or special wards [36]. The results of these previous studies are consistently supported in this study. A common result is that the higher the age above 60 years, the higher the proportion of adverse events and sentinel events, while the proportion of adverse and sentinel events reported by health providers was higher than that reported by patient safety officers. Currently, Korea has established patient safety committees at hospital-level medical institutions with more than 200 beds, and it is stipulated that dedicated patient safety personnel be assigned [4].

However, compared to the sizes of the hospitals, the number of dedicated personnel who are in charge of patient safety and perform overall patient safety activities is small. In this study, given the result of an increased probability of adverse or sentinel events being reported by medical personnel who are in close contact with patients and nurse them, it is necessary to form a culture in which all medical personnel in medical institutions participate in patient safety activities and report voluntarily. In addition, based on the assertion that patients and guardians should actively participate in patient safety activities as the best way to reduce medical accidents [37], patients and guardians are also active agents in securing and maintaining their own safety.

Among the types of PSIs, the ORs of the others type of events being classified as adverse events (3.81 times) and sentinel events (25.49 times) were higher than the OR of fall events. In this study, PSIs that belong to the others type refer to patient identification errors, self-extubation (tracheal intubation or drainage tube, etc.), suicide/self-harm, discharge/disappearance, violence/violence, burns, and bedsores. Events classified as others were 25.49 times more likely to be classified as sentinel events than medication/transfusion, suggesting that PSIs belonging to others could have fatal consequences for patients. Through future research, it is necessary to classify the events in the others type category that have the most influence on sentinel events, and to analyze them in detail.

In addition, although the incidence of equipment/computational disorder was small in this study, the OR of the incidents being classified as sentinel events was 4.01 times higher, requiring caution. Therefore, when performing medical treatment on or nursing a patient, equipment should be checked in advance and care should be taken to prevent computational errors.

Since this study extracted and analyzed data from medical institutions with more than 200 beds from the Korea Patient Safety Report data survey, this constitutes a limitation in not being able to analyze the PSIs of small hospitals with less than 200 people. In addition, since the analysis of the time of occurrence of incidents was classified into the working hours and seasons of nurses, it is necessary to explore the PSI related factors through a detailed analysis by month and time. Furthermore, healthcare providers may plan a multifaceted strategy to prevent PSIs by collecting information on the work environment, awareness of patient safety, and culture on the organizational systems of medical institutions.

## 5. Conclusions

This study was conducted to examine the factors related to domestic PSIs by analyzing the 2017, 2018, and 2019 KOPS data published by the Korea Institute for Healthcare Accreditation. In the three-year change in the PSIs, the proportion of adverse events and sentinel events decreased, while that of the near misses increased. Among the types of PSIs, falls, blood transfusions/medication, and others showed a high frequency. In this study, factors with a high probability of precipitating adverse events, based on near misses, were patient age, the seasons when PSIs occurred, reporter, hospital size, location, accident type, and medical department. In addition, factors with a high probability of causing sentinel events, based on near misses, were patient sex, patient age, incident reporter, the type of PSIs, and medical department. Therefore, to prevent sentinel events in PSIs, careful attention is required for older patients and female patients; it is necessary to understand and approach the complexity of internal medical diseases. In addition, compared to the medication/transfusion performed mainly by nurses in PSIs, falls, infection/contamination, equipment/computational failure, and surgery/anesthesia/inspection can lead to fatal results. Voluntary participation of medical personnel is required. Furthermore, it is necessary to establish a patient safety reporting system in which patients and their guardians can actively participate in patient safety activities and voluntarily report on PSIs.

## Figures and Tables

**Figure 1 ijerph-18-08482-f001:**
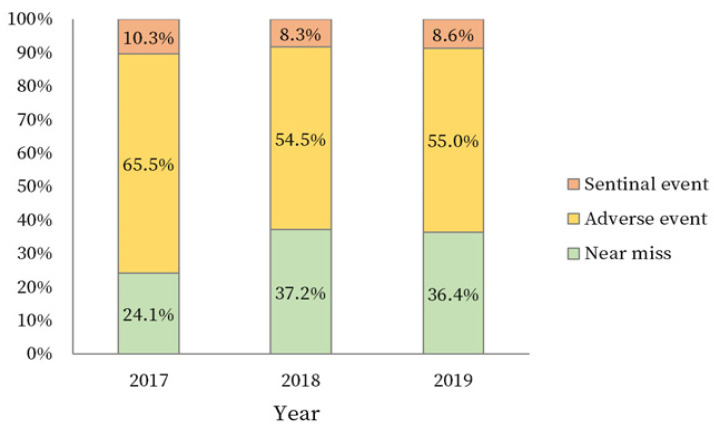
Three-year trend by severity of PSIs.

**Table 1 ijerph-18-08482-t001:** Difference in severity of PSIs by research variables.

Variables	Categories	Total (16,215)	Near Miss (*n* = 5588)	Adverse Event (*n* = 9201)	Sentinel Event (*n* = 1426)	χ^2^	*p*
		*n* (%)	*n* (%)	*n* (%)	*n* (%)		
Patient sex	Male	8067 (49.8)	2838 (35.2)	4697 (58.2)	532 (6.6)	96.93	<0.001
	Female	8148 (50.2)	2750 (33.8)	4504 (55.3)	894 (11.0)		
Patient age (years)	≤19	889 (5.5)	405 (45.6)	466 (52.4)	18 (2.0)	230.67	<0.001
	20–59	4293 (26.5)	1684 (39.2)	2372 (55.3)	237 (5.5)		
	≥60	11,033 (68.0)	3499 (31.7)	6363 (57.7)	1171 (10.6)		
Time when the PSI occurred	Day	7931 (48.9)	2907 (36.7)	4395 (55.4)	629 (7.9)	66.77	<0.001
	Evening	4278 (26.4)	1491 (34.9)	2411 (56.4)	376 (8.8)		
	Night	4006 (24.7)	1190 (29.7)	2395 (59.8)	421 (10.5)		
Seasons when the PSIs occurred	Spring	4065 (25.1)	1368 (33.7)	2328 (57.3)	369 (9.1)	6.75	0.345
	Summer	4642 (28.6)	1597 (34.4)	2656 (57.2)	389 (8.4)		
	Fall	3935 (24.3)	1345 (34.2)	2249 (57.2)	341 (8.7)		
	Winter	3573 (22.0)	1278 (35.8)	1968 (55.1)	327 (9.2)		
Incident reporter	Health provider	1523 (9.4)	419 (27.5)	898 (59.0)	206 (13.5)	67.81	<0.001
	Patient safety officer	14,692 (90.6)	5169 (35.2)	8303 (56.5)	1220 (8.3)		
Hospital size (beds)	200–499	8118 (50.1)	2707 (33.3)	4556 (56.1)	855 (10.5)	62.81	<0.001
	≥500	8097 (49.9)	2881 (35.6)	4645 (57.4)	571 (7.1)		
Location of occurrence of the PSIs	General ward	9078 (56.0)	3005 (33.1)	5253 (57.9)	820 (9.0)	133.21	<0.001
	Outpatient clinic	1864 (11.5)	771 (41.4)	1012 (54.3)	81 (4.3)		
	Special ward	1491 (9.2)	578 (38.8)	822 (55.1)	91 (6.1)		
	Others	3782 (23.3)	1234 (32.6)	2114 (55.9)	434 (11.5)		
Type of PSIs	Falls	9141 (56.4)	2554 (27.9)	5586 (61.1)	1001 (11.0)	1203.62	<0.001
	MT	2913 (18.0)	1524 (52.3)	1346 (46.2)	43 (1.5)		
	SAE	1591 (9.8)	835 (52.5)	689 (43.3)	67 (4.2)		
	IC	456 (2.8)	187 (41.0)	259 (56.8)	10 (2.2)		
	ECD	160 (1.0)	85 (53.1)	65 (40.6)	10 (6.3)		
	Others	1954 (12.1)	403 (20.6)	1256 (64.3)	295 (15.1)		
Medical department	Internal medicine	9596 (59.2)	3053 (31.8)	5566 (58.0)	977 (10.2)	136.74	<0.001
	Surgical	4923 (30.4)	1814 (36.8)	2737 (55.6)	372 (7.6)		
	Complex support division	1097 (6.8)	484 (44.1)	560 (51.0)	53 (4.8)		
	Clinical support division	599 (3.7)	237 (39.6)	338 (56.4)	24 (4.0)		

PSIs = patient safety incidents; MT = medication/transfusion; SAE = surgery/anesthesia/examination; IC = infection/contamination; and ECD = equipment/computational disorder.

**Table 2 ijerph-18-08482-t002:** Multinomial logistic regression analysis for predicting PSIs (*N* = 16,215).

Variables (Reference)	Categories	Adverse Event (Reference = Near Miss)			Sentinel Event (Reference = Near Miss)		
		Adjusted OR	95% CI	*p*	Adjusted OR	95% CI	*p*
Patient sex (Male)	Female	1.01	0.95–1.09	0.694	1.75	1.54–1.98	<0.001
Patient age (≤19 years)	20–59	1.23	1.05–1.43	0.010	2.97	1.80–4.90	<0.001
	≥60	1.35	1.17–1.57	<0.001	5.02	3.08–8.19	<0.001
Time when the PSI occurred (Day)	Evening	1.01	0.93–1.10	0.823	1.11	0.95–1.29	0.174
	Night	1.03	0.94–1.12	0.589	1.11	0.95–1.30	0.177
Seasons when the PSI occurred (Winter)	Spring	1.15	1.04–1.27	0.006	1.14	0.96–1.36	0.133
	Summer	1.12	1.01–1.23	0.029	1.03	0.86–1.22	0.776
	Fall	1.11	1.00–1.23	0.045	1.03	0.86–1.23	0.746
Incident Reporter (Patient safety officer)	Health provider	1.53	1.35–1.74	<0.001	2.23	1.85–2.70	<0.001
Hospital size (200∼499 beds)	≥500	1.22	1.13–1.31	<0.001	1.08	0.95–1.23	0.222
Location of the PSI (Others)	General ward	1.23	1.12–1.34	<0.001	1.01	0.87–1.16	0.918
	Outpatient clinic	1.54	1.32–1.78	<0.001	0.94	0.69–1.28	0.703
	Special ward	1.65	1.42–1.91	<0.001	1.40	1.05–1.87	0.021
Type of PSI (MT)	Falls	2.73	2.47–3.01	<0.001	11.83	8.58–16.31	<0.001
	SAE	0.75	0.65–0.86	<0.001	2.80	1.85–4.22	<0.001
	IC	1.64	1.34–2.02	<0.001	2.13	1.05–4.33	0.037
	ECD	0.76	0.54–1.06	0.101	4.01	1.93–8.34	<0.001
	Others	3.81	3.33–4.37	<0.001	25.49	18.10–35.90	<0.001
Medical department (Complex support division)	Internal medicine	1.42	1.23–1.63	<0.001	2.04	1.50–2.79	<0.001
	Surgical	1.26	1.09–1.46	0.002	1.45	1.05–2.00	0.023
	Clinical support division	1.92	1.53–2.42	<0.001	1.89	1.08–3.30	0.025

OR: odds ratio; CI: confidence interval; PSI: patient safety incident; TR = treatment room; MO = medical office; IR = injection room; ER = emergency room; ICU = intensive care unit; OR = operating room; RR = recovery room; MT = medication/transfusion; SAE = surgery/anesthesia/examination; IC = infection/contamination; and ECD = equipment/computational disorder.

## Data Availability

The data are available online at https://www.kops.or.kr/portal/board/stat/boardDetail.do?ctgryId=2&bbsId=stat&tmplatTyCode=J&nttNo=20000000002635, accessed on 21 February 2021. The KOPS data can be downloaded with permission from the Korea Institute for Healthcare Accreditation.

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
