# Peer review of "Trend Analysis of Patient Safety Incidents and Their Associated Factors in Korea Using National Patient Safety Report Data (2017~2019)"

_ijerph, 2021, doi:10.3390/ijerph18168482_

Round 1
Reviewer 1 Report
This is quite interesting paper concerning analyze trends in patient safety incidents (PSIs) and the factors associated with the PSIs by analyzing 2017–2019 Patient Safety Report Data in Korea. Authors analyzed probability of adverse events based on near misses and defined significant variables as patient age, the season when PSIs occurred, incident reporter, hospital size, the location of PSIs, the type of PSIs, and medical department. Additionally, the factors that were likely to precipitate sentinel events based on near misses were patient sex, patient age, incident reporter, the type of PSIs, and medical department.
The introduction clearly introduces the topic to be described, citing the relevant source publications. The results were collected and analyzed from a large amount of data.
The collected results are very extensive, but at the same time presented in a simple and legible way in figures and tables, enabling the reader to easily analyze these data. The presented discussion of the results and conclusions are accurate and correct.
My comments are the following:
- Introduction: some quotes are marked with a number in red and some in black.
- Arrange citations - on line 34 are quotations [16] instead of [5]. Reference no [5] in line 36 sholuld be [6] et cetera…
- Table 1 - The entire table should be on one page, together with the explanations of the abbreviations that appear in it.
In conclusion, this manuscript needs some minor corrections, before publishing in International Journal of Environmental Research and Public Health
Reviewer 2 Report
This paper examines the trend and associated factors with patient safety incidents in South Korea. I think it raises several interesting research points. However, the authors need to consider applying a relevant theoretical framework or fortify the introduction section with a literature review. Here are some detailed comments.
Abstract: What the standardized J-T statistics was at -8.28 mean? I hope the authors could add a brief explanation to this.
Introduction: The authors could provide why they focus their research on South Korea. Are there any distinct characteristics that South Korea (or hospitals in South Korea) have?
Materials and Methods: P 3, lines 108-110: The authors should define an adverse event and near miss here in addition to the sentinal event. Consider relocating the paragraph beginning "A near miss refers to~ (p.6) to p.3. The author should include what kind of demographic characteristics were controlled (consider including "measures" section).
I think the subsection "Discussion" has been missed on page 6, line 204.
Reviewer 3 Report
Introduction
47-49: The researchers speak of accidents due to medication, but in the results they present, they speak of falls as the highest incidence rate. It would be necessary to contextualize the association between these two closely related terms.
55-66: The researchers refer to the importance of patient safety reporting and the importance of patient safety reports but what reporting system is in place, who does the relevant reporting and what legislation is in place in the Korean population?
A discussion section should appear after the data analysis.
Conclusions
301: Researchers talk about a classification based on the flag system but this classification has not been used in the study variables or contextualized.
This is a fundamentally descriptive study that does not contribute relevant or differentiating data to the existing literature on the subject of patient safety.
It would be interesting to carry out studies on this subject that would provide protocols for action as well as tools for data collection.
The OMS has recently published a guide on patient safety that covers many of the aspects discussed in this paper.
Round 2
Reviewer 3 Report
The authors have made good modifications in response to the recommendations of the reviewers. The manuscript has improved.
It would be interesting to carry out studies on this subject that would provide protocols for action as well as tools for data collection.